# Molecular Background and Disease Prevalence of Biotinidase Deficiency in a Polish Population—Data Based on the National Newborn Screening Programme

**DOI:** 10.3390/genes13050802

**Published:** 2022-04-29

**Authors:** Aleksandra Jezela-Stanek, Lidia Suchoń, Agnieszka Sobczyńska-Tomaszewska, Kamila Czerska, Katarzyna Kuśmierska, Joanna Taybert, Mariusz Ołtarzewski, Jolanta Sykut-Cegielska

**Affiliations:** 1Department of Genetics and Clinical Immunology, National Institute of Tuberculosis and Lung Diseases, 01-138 Warsaw, Poland; 2Department of Inborn Errors of Metabolism and Paediatric, Institute of Mother and Child, 01-211 Warsaw, Poland; lidia.suchon@imid.med.pl (L.S.); joanna.taybert@imid.med.pl (J.T.); jolanta.cegielska@imid.med.pl (J.S.-C.); 3MedGen Medical Centre, 02-954 Warsaw, Poland; agnieszka.sobczynska@medgen.pl (A.S.-T.); kamila.czerska@medgen.pl (K.C.); 4Department of Newborn Screening and Metabolic Diagnostics, Institute of Mother and Child, 01-211 Warsaw, Poland; katarzyna.kusmierska@imid.med.pl (K.K.); mariusz.oltarzewski@imid.med.pl (M.O.)

**Keywords:** biotinidase deficiency, molecular background, newborn screening

## Abstract

Biotinidase deficiency (BD) is a rare autosomal recessive metabolic disease. Previously the disease was identified only by clinical signs and symptoms, and since recently, it has been included in newborn screening programs (NBS) worldwide, though not commonly. In Europe, BD prevalence varies highly among different countries, e.g., from 1:7 116 in Turkey to 1:75 842 in Switzerland. This paper aimed to present the molecular spectrum of BD (profound and partial forms) in Polish patients diagnosed within the national NBS of 1,071,463 newborns. The initial suspicion of BD was based on an abnormal biotinidase activity result determined in a dry blood spot (DBS) by colorimetric and by fluorimetric methods while biochemical verification was determined by serum biotinidase activity (as quantitative analysis). The final diagnosis of BD was established by serum enzyme activity and the *BTD* gene direct sequencing. The obtained results allowed for the estimation of disease prevalence (1:66,966 births, while 1:178,577 for profound and 1:107,146 for partial forms), and gave novel data on the molecular etiology of BD.

## 1. Introduction

Biotinidase is an enzyme responsible for releasing biotin from proteins [1]. Biotinidase deficiency (BD, MIM #253260) is an inborn error of biotin metabolism inherited by an autosomal recessive trait. BD results in multiple carboxylase deficiency (MCD), involved in the metabolism of carbohydrates, fats, and proteins. Two forms of MCD deficiency are distinguishable: early-onset form (holocarboxylase deficiency) and late-onset form (BD) [2].

Another classification of the disease is based on enzymatic activity: profound BD is recognized when biotinidase activity is below 10% of normal serum activity, partial deficiency—when it falls between 10 and 30% of normal serum activity [3].

Untreated patients suffer mainly from severe neurological symptoms such as hypotonia, seizures, ataxia, developmental delay, sensorineural deafness, and optic nerve atrophy [4,5]. Alopecia and skin rash (though not specific) are also frequently observed [4,5,6]. Treatment of BD consists of regular oral supplementation of biotin in pharmacological doses [7].

Previously BD was detected only by selective screening, i.e., the typical clinical signs [4]. Due to the limited knowledge of this disease among physicians (of various specializations such as pediatricians, neurologists, dermatologists, laryngologists, ophthalmologists, or allergologists) and its heterogeneous clinical presentation (e.g., may resemble skin allergy), many patients were not diagnosed or were misdiagnosed [7,8,9,10]. Delay in the diagnosis of BD results in irreversible clinical symptoms (mainly neurological ones) and even death with time, resulting from a lack of targeted treatment and administration of pharmacological doses of biotin [11].

To improve prognosis in patients with BD, recently, it has been added to the diseases panel of the newborn screening programs (NBS) in several countries (including Poland) [12,13]. The obtained data allow us to present the characteristics of the disease molecular background and its prevalence in the Polish population.

## 2. Materials and Methods

The study group was children with abnormal biotinidase activity results detected in the pilot newborn screening for biotinidase deficiency carried out in the Department of Screening and Metabolic Diagnostics (Institute of Mother and Child, Warsaw, Poland) from March 2016 to August 2020. During this period, 1,071,463 newborns were screened for biotinidase deficiency. The initial suspicion of BD was based on an abnormal biotinidase activity result determined in a dry blood spot (DBS)—by the colorimetric method from March 2016 to May 2018, and by the fluorimetric method from January 2018. Biochemical verification of suspected BD was determined by serum biotinidase activity (as quantitative analysis), based on the biotinidase-mediated breakdown reaction of N-biotinyl-p-aminobenzoate [14].

The final diagnosis of BD was established by serum enzyme activity and the *BTD* gene direct sequencing. 

Blood spots were collected on Whatman 903 screening card, and DNA was evaluated for genetic examinations from two 3 mm diameter discs using the Extract Blood PCR Kit (Sigma Aldrich, Merck, Darmstadt, Germany). Sanger sequencing of the *BTD* gene was performed for the patient and their family members (parents and siblings). PCR reaction mix and conditions are available on request. PCR products were purified using EXOSAP-IT (Amersham Biosciences, Amersham, UK) in accordance with the manufacturer’s protocol. Standard sequencing reaction and capillary electrophoresis using an ABI PRISM 3730 sequencer (ThermoFisher Science, Waltham, MA, USA) were performed, respectively. Fluorochromatograms were analyzed using Mutation Surveyor™ software (SoftGenetics, State College, PA, USA). Sequences NM_000060.3 NP_000051.1 were used for reference. Novel mutations were analyzed with the prediction module of the Alamut software. The presence of the gene variants in control populations was verified in the following databases: 1000 Genomes [15], the Exome Variant Server [16], the Exome Aggregation Consortium [17], gnomAD [18], and in own laboratory MedGen Medical Centre database containing Polish population data.

The control group consisted of 1097 patients from MedGen Laboratory Center’s database screened for pathogenic variants c.[1330G>C] (p.Asp444His; D444H) and c.[643C>T] (p.Leu215Phe; L215F) in the *BTD* gene.

## 3. Results

Of the 1 071 463 screened newborns, 16 cases were biochemically confirmed, 15 cases were genetically confirmed, including six profound and 9 partial deficiencies. 

Thirty-six pathogenic or likely pathogenic variants were found in 32 alleles (Table 1) and encompassed 15 different *BTD* variants. Among them (at the time of genetic testing), three new potentially pathogenic variants: p.Gly310Arg, c.309+2C>G, and p.Phe484Leu, were identified. In the remaining cases examined, variants previously described in the literature were detected: p.Asp444His, p.Leu215Phe, p.Gln456His, p.Cys418Ser, p.Cys33PhefsTer36, p.Gly488Asp, p.Thr532Met, p.Leu405Pro, p.Ala171Thr, p.Ile248Thr, p.Tyr438Cys, and p. Tyr540Cys (Table 2).

The molecular results are presented in Table 1.

The Estimated Prevalence of the Pathogenic Variants c.[1330G>C] (p.Asp444His; D444H) and c.[643C>T] (p.Leu215Phe; L215F) in the Control Group (Polish Population).

The p.Asp444His variant is located in exon 4 and results from a G1330>C transversion that changes histidine to aspartic acid at position 444 (p.Asp444His; D444H) [19]. 

Molecular analysis with next-generation sequencing performed in a group of 1097 individuals (non-affected with biotinidase deficiency) led to the identification of the pathogenic variant c.[1330G>C] (p.Asp444His; D444H) in 113 individuals (112 subjects had this variant in one allele, one subject had this pathogenic variant in both alleles in a homozygous state). These results allowed us to estimate the prevalence of the p.Asp444His variant in the study group as 10.3%.

The frequency of this variant among the alleles studied (no. 2194) is 5.2%.

The pathogenic variant L215F was identified in one patient among 1097 controls. It gave the frequency of the L215F variant in the study group as 0.09% 

The frequency of this variant among the tested alleles (no. 2194) is 0.045%.

## 4. Discussion

Biotinidase deficiency is a form of multiple carboxylase deficiency caused by homozygous/compound heterozygous mutations in the *BTD* gene (609019) on chromosome 3p25 composed of 4 exons and 3 introns [20]. So far, according to the HGMD database, 252 pathogenic variants in the *BTD* gene have been registered [21]. In the *BTD* database, 221 variants were registered, among them: 77% missense, 11% deletion, 7% nonsense, 3% insertion, 2% duplication, 1% silent [22]. The most frequently detected variants are: p.Asp444His, p.Cys33PhefsTer36, p.Gln456His, p.Arg538Cys, and p.Ala171Thr+p.Asp444His [7]. The changes mentioned above occur in 60% of the alleles tested [23].

The performed, presented herein tests, including molecular data in correlation with serum enzyme activity among Polish patients, allowed us to establish the final diagnosis of profound biotinidase deficiency in 6 cases and partial deficiency in 10 patients.

### 4.1. Molecular Characteristics of Polish BD Individuals

All but two (c.98_104delinsTCC and c.184G>A) identified *BTD* variants were localized in exon 4. Contrary to other mutations, which were missense changes, c.98_104delinsTCC is indel.

The p.Asp444His variant was most frequently identified in the study group and accounted for 33% of detected mutations. The second most frequent was p.Leu215Phe, in 20% of pathogenic variants detected. The third in terms of frequency was c.309+2C>G and c.98_104delinsTCC (5.5%). The remaining mutations were noted in single patients and accounted for 32% of all variants identified in this group (Table 2).

The p.Asp444His *BTD* variant is frequent in the European population. As one can find in the GnomAD/Genomes+Exomes, this allele frequency is 0.031839%. Thus, it may be responsible for some of the variation in enzyme activity observed among individuals with low-normal serum biotinidase activity [24].

### 4.2. Novel Variants of the BTD Gene

Two of three novel variants identified in the Polish cohort—p.Gly310Arg and c.309+2C>G—have been placed in the cis configuration with a known variant p.Asp444His as a complex allele, not previously reported in databases. In the case of Patient 12, the child’s parents were also examined, and it was found that the father carries the pathogenic variant p.Ala171Thr+p.Asp444His and the mother has the potentially pathogenic variant p.Gly310Arg+p.Asp444His.

The c.309+2C>G variant is classified as likely pathogenic mutation (ACMG criteria: PVS1_Strong + PM2 + PM3_Supporting); c.1452T>G (p.Phe484Leu) as VUS (variant of unknown significance), based on ACMG criteria: PM2 + PM3.

### 4.3. Genotype-Phenotype Correlation

The definite correlation has been possible only in two patients, homozygotes for c.[643C>T];[643C>T] (p.[Leu215Ph];[Leu215Phe]; Patient 11) and c.[309+2C>G;1330G>C]; [309+2C>G;1330G>C] (p.[?;Asp444His]; [?Asp444His]; Patient 13) variants. However, the genotype of the latter patient (13) is more complex; thus, the clear correlation needs further verification, based on other clinical reports. Of the rest, 12 individuals were compound heterozygotes which makes it very difficult to infer the significance and association of a single variant with the phenotype; in one, a mutation was identified only on one allele of the *BTD* gene—pathogenic variants c. 643C>T. However, given the reduced biotinidase activity, which corresponds to a diagnosis of partial biotinidase deficiency, the proband was classified as *BTD* affected with *BTD* deficiency.

### 4.4. Clinical Characteristics of BTD Known Variants Identified during Polish NBS with Literature Review

#### 4.4.1. p.Asp444His

According to the literature, almost all individuals with partial biotinidase deficiency have the pathogenic variant p.Asp444His in one allele of *BTD* in combination with a pathogenic variant for profound deficiency in the other allele [19].

In the study group, the most frequently identified variant was p.Asp444His, located in exon 4 and arising from a G1330>C transversion that results in a change from histidine to aspartic acid at position 444 (p.Asp444His; D444H) [19]. It has been shown that the p.Asp444His variant in one allele in combination with a pathogenic variant for profound biotinidase deficiency in the other allele is the most common cause of partial biotinidase deficiency and results in enzyme activity in the range of about 20–25% [19], which is confirmed by the results of studies of Polish patients. 

It is reported that p.Asp444His in a homozygous state can result in a biotinidase activity of about 50%, which does not require treatment [19]. However, in light of various publications, this information is not consistent. Patients with this pattern of pathogenic variants may show reduced enzyme activity that mandates a diagnosis of partial biotinidase deficiency [25,26]. Moreover, symptomatic patients with the p.Asp444His/p.Asp444His genotype have also been reported [27]. Symptoms included developmental delay, autism spectrum disorders, hearing loss, and skin lesions, and the time of clinical manifestation of the disease ranged from 2 years to 6.5 years [26]. In addition, individuals homozygous for the p.Asp444His variant have been reported, who had enzyme activity that excluded the diagnosis of biotinidase deficiency, but who developed symptoms of the disease—skin lesions (severe diaper dermatitis, seborrheic dermatitis) [28]. Nevertheless, most individuals with the described gene variant in the homozygous state remain asymptomatic [7,26]. On the other hand, the estimated prevalence of this p.Asp444His change for the general population is 3% to 4% [24]; in Hungarian people, it was calculated as 5.5% [29], while in the homozygous state is 0.15% [30]. Interestingly, in the Polish population, the estimated c.[1330G>C] (p.Asp444His) prevalence seems to be relatively high, i.e., 10.3%.

#### 4.4.2. p.Leu215Phe

The p.Leu215Phe variant mapping to exon 4 was the second most frequent in the studied group. It results in a profound enzyme deficit when another pathogenic variant is present in the second allele or when p.Leu215Phe is present in a homozygous trait [24,31].

Usually, the p.Asp444His variant was present in the second allele, resulting in a partial biotinidase deficiency, and patients were asymptomatic [31,32,33]. Few cases of patients with this variant were found in the literature. We found a description of a patient with a late-onset biotinidase deficit, diagnosed symptomatically, who presented with optic nerve atrophy and tetraplegia. The patient carried two pathogenic variants in the L215F/R538C heterozygous system, resulting in a profound enzyme deficit [34].

In a heterozygote carrying c.[1330G>C] and another variant on the second allele, the enzyme activity corresponds to a partial biotinidase deficiency [19]. Eleven patients with this variant have been detected in the Dutch population in newborn screening. There is no information if any of them had clinical symptoms of the disease; biotinidase activity in this group ranged from 18% to 30% [35].

The Polish literature describes patients diagnosed by symptomatic screening with the L215F variant in the homozygous state—two with partial (10–30%), one with profound (<10%) BT activity; in the remaining cases, no data on enzyme activity has been presented [36]. Three of the four patients had optic nerve atrophy, and two of them were diagnosed with hearing loss. These individuals were from the northeastern part of the country.

Patients observed in IMID with the L215F variant in one or both alleles were from the central-eastern part of Poland. Due to the high detection rate of this variant in Polish patients, the prevalence of this variant was determined in the control group, which appeared to be low, i.e., 0.09%. Significantly, no other country has reported a frequency in the general population for this variant.

#### 4.4.3. p.Gln456His

The pathogenic variant p.Gln456His was first described in 1997 by Norrgard [24]. It is one of the most common variants causing profound biotinidase deficiency [7]. Its high detection rate in the Hungarian population has been reported, and its prevalence has been determined to be 0.5% [29]. In the homozygous state, the variant causes, as mentioned, a profound biotinidase deficiency [24,37]. 

A symptomatic patient with a profound enzyme deficit, however, presented no symptoms typical for BD (hepatomegaly only was noted), being homozygote for this variant, has been described in the literature, diagnosed at the age of 10 days [37]. In addition, symptomatic patients with partial enzyme deficiency and the Q456H/c.[1330G>C] variants have been reported with hypertonia, skin lesions, and hair loss. The clinical onset was noted between 4 and 6 months of age [37,38]. A Polish patient with this genotype (Table 2) is on a diet treatment thus remained asymptomatic to date.

#### 4.4.4. p.Cys33PhefsTer36

p.Cys33PhefsTer36 is a variant resulting from a reading frame change and premature termination of protein synthesis in exon 2 due to the deletion of the coding sequence from G at position 98 to G at position 104 and insertion of the TCC sequence [39]. This change results in a profound biotinidase deficiency [23,31,38], the enzymatic activity of patients who are homozygotes of the p.Cys33PhefsTer36 variant is close to 0% [28,40,41,42]. Eight Hungarian patients with the p.Cys33PhefsTer36/p.Asp444His genotype have been shown with enzyme activity between 10% and 17% [33]. The biotinidase activity of a German and a Hungarian patient (separate publication) with the same genotype was 9% and 9.2%, respectively [37,43]. Patients with the p.Asp444His/Cys33PhefsTer36 variant configuration showed activity ranging from 18.6% to 30.1% [19]. 

A case of a symptomatic patient diagnosed with profound biotinidase deficiency who was a p.Cys33PhefsTer36/Cys33PhefsTer36 homozygote has been described [41]. Neurological abnormalities—including hearing loss and central nervous system lesions—predominated among the symptoms that appeared at 15 months of age. Other patients with this genotype who developed encephalopathy and seizures have also been reported [42]. Patients with profound biotinidase deficiency carrying pathogenic variants (one of which was p.Cys33PhefsTer36) have been reported in the literature. A patient with the p.Cys33PhefsTer36/p.P498Ffs*13 genotype was symptomatic at two months of age and presented with drug-resistant seizures and skin lesions in the gluteal region [44]. Another patient, who was diagnosed with the pathogenic p.Ala171Thr+p.Asp444His/Cys33PhefsTer36 variant, is one of the few individuals who developed the symptomatic disease at an adult age. The patient was diagnosed at 36 years of age and presented with diplegia, lower limb pain, and vision loss [45]. In a patient diagnosed with partial biotinidase deficiency and genotype p.Cys33PhefsTer36/p.Asp444His, recurrent hypoglycemia (atypical sign in BD) and metabolic acidosis at 19 years of age have been reported [28]. Other patients with the same genotype showed hypotonia and skin lesions [38]. The p.Cys33PhefsTer36 variant is reported to be statistically significantly more common in symptomatic patients with biotinidase deficiency [23].

#### 4.4.5. p.Thr532Met

The p.Thr532Met variant was first described by Swango in 1998 [19]. Mutations near the carboxyl terminus of the protein have been found to abolish enzyme activity, but the pathomechanism of this process is not understood [46]. In a homozygous state, p.Thr532Met results in a profound enzyme deficiency [37,46]. Symptomatic patients with this variant have been discussed in the Hungarian population, and all individuals were from the Roma ethnic group [37]. The onset of symptoms ranged from 1.5 weeks to 7 weeks; the patients showed developmental delay and stridor. In the Turkish population, the variant has been reported in individuals with profound biotinidase deficiency who were p.Thr532Met homozygotes and presented with skin rashes, vision loss, and abnormal urinary organic acid profile [46].

Developmental delay, skin rashes, and vomiting during infection were observed among p.Thr532Met/p.Asp444His heterozygotes diagnosed with partial biotinidase deficiency [38]. Newborn screening in the Netherlands allowed detecting five patients carrying the p.Thr532Met/p.Asp444His variant. Three of them had an enzyme activity of 28%, and the others showed activity of 27% and 23% [35].

High detection of individuals carrying p.Thr532Met in one or both gene copies was found in the Turkish population, former Yugoslavia, Austrians of Turkish origin, and Roma living in Rome and Hungary [37,42,43,46]. In the Polish population, one patient homozygous for this variant diagnosed in symptomatic screening at the age of 14 months has been reported so far [36]. Therefore, this variant seems to be associated with relatively mild and/or possibly later clinical onset.

#### 4.4.6. p.Ala171Thr+p.Asp444His

The p.Ala171Thr+p.Asp444His complex variant results from a guanine to adenine substitution at position 511 near the 5′ end of exon 4, resulting in a threonine to alanine substitution at position 171 (A171T), and a guanine to cytosine substitution at position 1330 near the 3′ end of exon 4, resulting in a histidine to aspartic acid substitution at position 444. This mutation has been reported as inherited from the same parent as a double complex allele (in cis configuration) [24]. In a homozygous state, this variant results in a profound enzyme deficiency, similar to associated other pathogenic variants in the other allele [7,24]. Patients who are homozygotes for p.Ala171Thr+p.Asp444His have been described and treated with biotin; during an eight-year follow-up, they did not develop symptoms of BD [26]. 

A patient with the p.Ala171Thr+p.Asp444His/Cys33PhefsTer36 variant combination was described previously [45]. No symptomatic individuals with partial biotinidase deficiency who had the p.Ala171Thr+p.Asp444His complex variant (in cis) have been reported in the literature.

The occurrence of p.Ala171Thr+p.Asp444His in a homozygous state in patients with profound enzyme deficiency is common. Among the *BTD* alleles analyzed, this variant accounted for 17.3% in the USA [23], 3.1% in Austria [47], and 16.7% in Hungary [29]. 

#### 4.4.7. p.Cys418Ser

In a homozygous state, p.Cys418Ser causes a profound enzyme deficiency [42]. No symptomatic patients homozygous for this mutation have been described in the literature. The mutation was first described in the Hungarian population by László in 2003. [22,37]. The partial biotinidase deficiency is observed when the p.Asp444His variant is present in the second allele. The clinical status of patients with a pathogenic variant other than p.Asp444His in the second allele has not been clearly defined. In the case of one Polish proband with a p.Leu215Phe/p.Cys418Ser genotype, a partial enzyme deficit was diagnosed, but with borderline values—close to 10%. In the case of one patient with the p.Cys418Ser/p.Asp444His variant and serum enzyme activity of 24%, mild hypotonia was reported as a clinical manifestation of the disease [37]. Another patient with the combination of the same variant showed a similar serum enzyme activity of 24.5% [33].

#### 4.4.8. p.Gly488Asp

The p.Gly488Asp variant described by Pomponio in 1997 [48] has not been reported among symptomatic patients. Based on the presented Polish data, it can be concluded that Gly488Asp in configuration with p.Thr532Met results in a profound biotinidase deficiency.

#### 4.4.9. p.Leu405Pro

p.Leu405Pro is a pathogenic variant that was first described in a Swedish population [49]. To date, one patient with a partial biotinidase deficiency, being a compound heterozygote c.[1214T>C]/c.[1330G>C] (as in our Patient 1), has been described. The proband was detected by newborn screening and did not show any signs of disease [49].

#### 4.4.10. p.Tyr438Cys

The p.Tyr438Cys variant was first described in a Polish patient in a publication by Wolf in 2002 [50]. Together with the p.Cys33PhefsTer36 variant in the second allele (in trans) resulted in a profound biotinidase deficiency with a serum enzyme activity of 1%. In another publication, the same variants were reported also in a Polish patient, detected by screening, whose serum biotinidase activity was 0% [36]. The patient was treated with biotin and remained asymptomatic. 

#### 4.4.11. p.Gly310Arg

Another variant, p.Gly310Arg, has been reported to date only once [51]. The child presented with seizures observed from the 6 h after birth and non-bullous ichthyosiform erythroderma, patchy alopecia with sparse hair, oedematous fingers, and toes. Holocarboxylase deficiency was considered at first, and the authors concluded an unusual *BTD* deficiency presentation. 

#### 4.4.12. p.Tyr540Cys

Norrgard first described the p.Tyr540Cys variant in a proband detected as part of a newborn screening test in the United States [23]. No symptomatic patients who carry this pathogenic variant in one of their alleles are known to date [23].

#### 4.4.13. p.Ile248Thr

The p.Ile248Thr variant was first described in 2016 by Procter [52] in a pathogenic variant p.K176N on the second allele. The diagnosis of biotynidase deficiency was supported by decreased activity of the enzyme in blood serum.

## 5. Conclusions

With the presented results, we provided novel data on the molecular phenotype and clinical course of biotinidase deficiency among Polish patients. We performed analyses among 1,071,463 newborns, which is, to our knowledge, the largest European population screened for the disease. The data obtained have enabled us to propose the following conclusions.

Regarding the most frequently observed *BTD* variant—c.[1330G>C] (p.[Asp444His]), its clinical consequences depend on the variant on the second allele of the gene. The patients who are homozygous for the p.Asp444His pathogenic variant are expected to have serum biotinidase enzyme activity consistent with partial *BTD* deficiency, and thus may not require biotin therapy and should only be monitored or—if treated—low doses of biotin are recommended. Compound heterozygotes for the p.Asp444His pathogenic variant may have profound biotinidase deficiency but only in the presence of a pathogenic variant that results in profound serum biotinidase enzyme activity. Moreover, another interesting finding from our study, being of high clinical importance, is that this variant *in cis* with the novel one (c.309+2C>G), which makes these complex mutations, is disease causing. 

Contrarily, homozygote Leu215Phe presents with profound BT deficiency, while in a heterozygote state, this variant results in partial *BTD* deficiency. Considering that almost all children with profound *BTD* deficiency become symptomatic if not treated, molecular diagnostics is essential for family counseling and early assessment in subsequent pregnancy.

## Figures and Tables

**Table 1 genes-13-00802-t001:** Genotype characteristics and resulted biochemical phenotype of Polish BD cases.

Patient ID	Genotype (According to Nucleotide and Protein Reference Sequence)	Enzyme Activity— in DBS [%]/in Serum [%]	BD Form
1	c.[1330G>C];[1214T>C] p.[Asp444His]; Leu405Pro]	23/27	partial
2	c.[1330G>C];[1368A>C] p.[Asp444His];[Gln456His]	30/30	partial
3	c.[1330G>C];[643C>T] p.[Asp444His];[Leu215Phe]	23/22	partial
4	c.[643C>T];[=] p.[Leu215Phe];[=]	0.08/19	partial
5	c.[643C>T];[1253[G>C] p.[Leu215Phe];[Cys418Ser]	6.5/11	partial
6	c.[643C>T];[1330G>C] p.[Leu215Phe]; [Asp444His]	18.2/24	partial
7	genetic testing has not been performed	23/20	partial
8	c.[1330G>C];[1619A>G] p.[Asp444His];[Tyr540Cys]	24/16	partial
9	c.[98_104delinsTCC];[1330G>C] p.[Cys33PhefsTer36];[Asp444His]	18.3/23	partial
10	c.[643C>T];[1330G>C] p.[Leu215Phe];[Asp444His]	14.4/26	partial
11	c.[643C>T];[643C>T] p.[Leu215Ph];[Leu215Phe]	2.4/0	profound
12	c.[928G>A;1330G>C];[511G>A;1330G>C] p.[(Gly310Arg+p.Asp444His)]; [(Ala171Thr+p.Asp444His)]	2.9/0	profound
13	c.[309+2C>G;1330G>C]; [309+2C>G;1330G>C] p.[?;Asp444His];[?Asp444His]	10.2/0	profound
14	c.[1463G>A];[1595C>T] p.[Gly488Asp];[(Thr532Met)]	8.8/0	profound
15	c.[98_104delinsTCC];[1313A>G] p.[Cys33PhefsTer36];[Tyr438Cys]	2.8/0	profound
16	c.[743T>C]; [1330G>C];[1452T>G] p.[Ile248Thr]; [Asp444His];[Phe484Leu]	12.8/9	profound

DBS—dry blood spot; [=]—no pathogenic variant identified.

**Table 2 genes-13-00802-t002:** *BTD* pathogenic variants identified in Polish newborn screening (% of allele frequency).

Nucleotide Change	Exon	Variant Type	Amino Acid Change	Allele Frequency (%)
c.1330G>C	Exon 4	Missense	p.Asp444His	12/36 (33%)
c.643C>T	Exon 4	Missense	p.Leu215Phe	7/36 (20%)
c.98_104delinsTCC	Exon 2	Indel	p.C33Ffs*36 (Cys33PhefsTer36)	2/36 (5.5%)
c.309+2C>G (novel)	Exon 4	Missense	p.?	2/36 (5.5%)
c.1368A>C	Exon 4	Missense	p.Gln456His	1/36 (3%)
c.1253G>C	Exon 4	Missense	p.Cys418Ser	1/36
c.1463G>A	Exon 4	Missense	p.Gly488Asp	1/36
c.1214T>C	Exon 4	Missense	p.Leu405Pro	1/36
c.511G>A;	Exon 4	Missense	p.Ala171Th	1/36
c.1313A>G	Exon 4	Missense	p.Tyr438Cys	1/36
c.928G>A (novel)	Exon 4	Missense	p.Gly310Arg	1/36
c.1619A>G	Exon 4	Missense	p.Tyr540Cys	1/36
c.1452T>G (novel)	Exon 4	Missense	p.Phe484Leu	1/36
c.743T>C	Exon 4	Missense	p.Ile248Thr	1/36
c.1595C>T	Exon 4	Missense	p.Thr532Met	1/36

## Data Availability

The data presented in this study are available on request from the corresponding author.

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
