# Peer review of "Molecular Background and Disease Prevalence of Biotinidase Deficiency in a Polish Population—Data Based on the National Newborn Screening Programme"

_genes, 2022, doi:10.3390/genes13050802_

Round 1

Reviewer 1 Report

Line 92-93. Autors reported that, Thirty-six pathogenic or likely pathogenic variants were found in 32 alleles (Table 1) and encompassed 16 different BTD variants. In Table 2 and manuscript lines 96-97-98, only 15 different BTD variants are mentioned. This difference needs correction by the authors. 

Table 2. Allel frequency of p.Asp444His variant should be 12 according to the allels reported in Table 1. Therefore the information in Line 133-134. The p.Asp444His variant was most frequently identified in the study group and accounted for 39% of detected mutations, should be changed accordingly to 33%.

Lines 153-155. The definite correlation has been possible only in two patients, homozygotes for c.[1330G>C];[1330G>C] (p.[Asp444His];[Asp444His]) and c.[643C>T];[643C>T] (p.[Leu215Ph)];[Leu215Phe]) variants. Which patient is mentioned as homozygote for c.[1330G>C];[1330G>C] needs clarification. The authors also commented in discussion that homozygosity of this mutation is disputable for clinical significance. Patients 12 and 13 in the study with c.[1330G>C];[1330G>C] harbor other variants as well that changes the clinical significance of this mutation  for a definite correlation, when other variants are present. 

Lines 333-335. The patient who are homozygous for the p.Asp444His pathogenic variant are expected to have serum biotinidase enzyme activity consistent with partial BTD, thus may not require biotin therapy but should be monitored. Comment: The message here may be understood as no tretament but follow-up for partial BTD, however patients with partial BTD are recommended biotin treatment but with a lower dosage. The authors may consider changing this statement.

Author Response

Dear Reviewer,

thank you very much for your efforts, and time spent on our manuscript, but mostly for important suggestions and comments.

Accordingly, we have improved the article, as described below:

Line 92-93. Autors reported that, Thirty-six pathogenic or likely pathogenic variants were found in 32 alleles (Table 1) and encompassed 16 different BTD variants. In Table 2 and manuscript lines 96-97-98, only 15 different BTD variants are mentioned. This difference needs correction by the authors. 

thank you, it was definitely our mistake; now the information has been corrected

Table 2. Allel frequency of p.Asp444His variant should be 12 according to the allels reported in Table 1. Therefore the information in Line 133-134. The p.Asp444His variant was most frequently identified in the study group and accounted for 39% of detected mutations, should be changed accordingly to 33%.

as above -   thank you very much for this comment, it was definitely our mistake; now the information has been corrected

Lines 153-155. The definite correlation has been possible only in two patients, homozygotes for c.[1330G>C];[1330G>C] (p.[Asp444His];[Asp444His]) and c.[643C>T];[643C>T] (p.[Leu215Ph)];[Leu215Phe]) variants. Which patient is mentioned as homozygote for c.[1330G>C];[1330G>C] needs clarification. The authors also commented in discussion that homozygosity of this mutation is disputable for clinical significance. Patients 12 and 13 in the study with c.[1330G>C];[1330G>C] harbor other variants as well that changes the clinical significance of this mutation  for a definite correlation, when other variants are present. 

thank you so much for these comments; we have rewritten the fragment accordingly, changed the variants and added probands ID; now the sentences are as follows:

The definite correlation has been possible only in two patients, homozygotes for c.[643C>T];[643C>T] (p.[Leu215Ph];[Leu215Phe]; Patient 11) and c.[309+2C>G;1330G>C];[309+2C>G;1330G>C] (p.[?;Asp444His];[?Asp444His]; Patient 13) variants. However, the genotype of the latter patient (13) is more complex thus the clear correlation needs further verification, based on another clinical reports. Of the rest…

Lines 333-335. The patient who are homozygous for the p.Asp444His pathogenic variant are expected to have serum biotinidase enzyme activity consistent with partial BTD, thus may not require biotin therapy but should be monitored. Comment: The message here may be understood as no tretament but follow-up for partial BTD, however patients with partial BTD are recommended biotin treatment but with a lower dosage. The authors may consider changing this statement.

thank you very much for this very valuable remark. Indeed, the message may be misunderstood, and we have changed the sentence for:

The patients who are homozygous for the p.Asp444His pathogenic variant are expected to have serum biotinidase enzyme activity consistent with partial BTD, thus may not require biotin therapy and should only be monitored or – if treated – low doses of biotin are recommended

Reviewer 2 Report

The authors presented molecular results along with clinical phenotypes from a pilot study of BD deficiency among Polish neonates using newborn dried blood spot molecular testing.   Three novel variants were described among other known variants.
The data reported adds to the growing literature concerning genetic variants that are potentially BD causing.   The authors supply evidence to include a molecular panel as a second tier after laboratory screening for BD at birth.  The data is strong and the research is valuable.

Abstract:  The abstract does not represent the study well.  It lacks methods and a sentence about the actual molecular findings.

Editing:  The writing can be improved by careful copy-editing.  Some examples:  

line-106  p.Asp444His variant is located in exon 4 and result from a G1330>C transversion that consequences results in a change of histidine to aspartic acid at position 444 (D444H) [19].

line-200;  There is no information that if   any of them had clinical symptoms of the disease;

line-218: A symptomatic patient with a profound enzyme deficit, however presenteding with no ( symptoms) typical for BD symptoms

Body. The authors should choose a single nomenclature to describe the variants (c.DNA or p.DNA), and indicate the the other nomenclature or legacy name in brackets following the chosen name, when appropriate. Example: p.ASp444His switching with c.1330G and D444H.   I found this switching back and forth in the text to be confusing, which detracted from the ready absorption of the material.  However the authors did adhere to the c.DNA name for section 4.3  lines 153-155, but should do this througout.

Table I.  The content is good but needs a footnote or legend to describe what the table is telling the reader.  What does [=] mean in case number 4?  The nucleotide protein names could be in separate columns to make comparisons easier.  Also, the authors could organize the table by listing the more common variant first (or the know pathogenic mutation first)  and this may help the reader interpret the data more easily. 

  Table II.  Include a column with known pathogenicity,  if available. 

Content.  From the discussion and results it appears that p.Asp444His is common and of varying clinical consequence.  The most interesting finding is that this variant in cis with the novel variants  which makes these complex mutations indeed disease-causing.  This is an important observation even though the authors can not isolate the second mutation in terms of clinical consequence.  This finding should be emphasized.

The authors are thorough in their discussion of the other case findings but discussion of the other mutations could be better organized--The key here is to organize the discussion around Asp444His perhaps even though it is not strictly disease causing on its own.  The authors may find a better way to organize and summarize this, other than as a case series.

There is potential for an excellent manuscript with better organization and editing.

Author Response

Dear Reviewer,

thank you very much for your efforts, and time spent on our manuscript, but mostly for important suggestions and comments.

Accordingly, we have improved the article, as described below:

Abstract:  The abstract does not represent the study well.  It lacks methods and a sentence about the actual molecular findings.

thank you for the remark; now the information has been added and the Abstract includes:

This paper aimed to present the molecular spectrum of BD (profound and partial forms) in Polish patients diagnosed within the national NBS of 1 071 463 newborns. The initial suspicion of BD was based on an abnormal biotinidase activity result determined in a dry blood spot (DBS) by colorimetric and by fluorimetric methods while biochemical verification was determined by serum biotinidase activity (as quantitative analysis). The final diagnosis of BD was established by serum enzyme activity and the BTD gene direct sequencing. The obtained results allowed for…

Editing:  The writing can be improved by careful copy-editing.  Some examples:  

we are very grateful for these suggestions; now the text has been improved and we have corrected as listed below

line-106  p.Asp444His variant is located in exon 4 and result from a G1330>C transversion that consequences results in a change of histidine to aspartic acid at position 444 (D444H) [19].

is now: p.Asp444His variant is located in exon 4 and results from a G1330>C transversion that changes of histidine to aspartic acid at position 444 (p.Asp444His ; D444H) [19].

line-200; There is no information that if   any of them had clinical symptoms of the disease;

is now: There is no information if any of them had clinical symptoms of the disease; biotinidase activity in this group ranged from 18% to 30% [35].

line-218: A symptomatic patient with a profound enzyme deficit, however presenteding with no (symptoms) typical for BD symptoms

is now: A symptomatic patient with a profound enzyme deficit, however presented no symptoms typical for BD (hepatomegaly only was noted),

Body. The authors should choose a single nomenclature to describe the variants (c.DNA or p.DNA), and indicate the the other nomenclature or legacy name in brackets following the chosen name, when appropriate. Example: p.ASp444His switching with c.1330G and D444H.   I found this switching back and forth in the text to be confusing, which detracted from the ready absorption of the material.  However the authors did adhere to the c.DNA name for section 4.3  lines 153-155, but should do this througout.

yes, you are absolutely right that the nomenclature was confusing; now we have made corrections throughout the manuscript, i.a.:

line 90-93: The control group consisted of 1 097 patients from MedGen Laboratory Center's database screened for pathogenic variants c.[1330G>C] (p.Asp444His; D444H) and c.[643C>T] (p.Leu215Phe; L215F) in the BTD gene.

Table 2.

Table I.  The content is good but needs a footnote or legend to describe what the table is telling the reader.  What does [=] mean in case number 4?  The nucleotide protein names could be in separate columns to make comparisons easier.  Also, the authors could organize the table by listing the more common variant first (or the know pathogenic mutation first)  and this may help the reader interpret the data more easily. 

thank you for these suggestions; we have rephrased the title and separated the nucleotide and protein nomenclature;

we have organized the table based on partial/profound enzyme deficiency and in the order in which they were diagnosed – if you don’t mind, we prefer to leave the table as it is now

Table II.  Include a column with known pathogenicity, if available. 

thank you, the information has been added

Content.  From the discussion and results it appears that p.Asp444His is common and of varying clinical consequence.  The most interesting finding is that this variant in cis with the novel variants which makes these complex mutations indeed disease-causing.  This is an important observation even though the authors can not isolate the second mutation in terms of clinical consequence.  This finding should be emphasized.

yes, you are definitely right and we like to thank you very much for this comment. The sentence has now been added:

Moreover, another interesting finding from our study, being of high clinical importance, is that this variant in cis with the novel one (c.309+2C>G), which makes these complex mutations, is disease-causing. 

The authors are thorough in their discussion of the other case findings but discussion of the other mutations could be better organized--The key here is to organize the discussion around Asp444His perhaps even though it is not strictly disease causing on its own.  The authors may find a better way to organize and summarize this, other than as a case series.

thank you; we like to explain that, one of our aims was to give the molecular background of BT deficiency thus we have summarized all the variants identified in the Polish population but – as most of them are really rare – concluded only on the two of them.

We do hope such an organization is rational from this perspective.

There is potential for an excellent manuscript with better organization and editing.

thank you, we agree and thus some reorganization has been made, we:

- rephrased the Tables’ titles

- checked and corrected variants' nomenclature

- added subtitles in the Discussion (separating the given variants).
